# *PafS* Containing GGDEF-Domain Regulates Life Activities of *Pseudomonas glycinae* MS82

**DOI:** 10.3390/microorganisms10122342

**Published:** 2022-11-26

**Authors:** Xianyi Chen, Shaoxuan Qu, Xin Luo, Shi-En Lu, Youzhou Liu, Huiping Li, Lijuan Hou, Jinsheng Lin, Ning Jiang, Lin Ma

**Affiliations:** 1Jiangsu Key Laboratory for Horticultural Crop Genetic Improvement, Institute of Vegetable Crops, Jiangsu Academy of Agricultural Sciences, Nanjing 210014, China; 2Department of Biochemistry, Molecular Biology, Entomology and Plant Pathology, Mississippi State University, Starkville, MS 39762, USA; 3Institute of Plant Protection, Jiangsu Academy of Agricultural Sciences, Nanjing 210014, China

**Keywords:** *Pseudomonas glycinae* MS82, GGDEF domain, *PafS*, life activities

## Abstract

Cyclic dimeric guanosine monophosphate (c-di-GMP) is synthesized by diguanylate cyclase (DGC) with the GGDEF domain. As a ubiquitous bacterial second messenger, it regulates diverse life-activity phenotypes in some bacteria. Although 38 genes encoding GGDEF-domain-containing proteins have been identified in the genome of the *Pseudomonas glycinae* strain MS82, whether c-di-GMP functions as a facilitator or repressor of life-activity phenotypes is poorly understood. In this study, one of the 38 genes containing a GGDEF domain in MS82, *PafS* was investigated to explore its regulatory function in bacterial life activities. The *PafS-*deletion mutant Δ*PafS* and reversion mutant *PafS*-comp were constructed by the method of biparental conjugation and homologous recombination. The life activities of the mutants, such as antifungal activity, biofilm formation ability, polysaccharide content, and motor behavior, were explored. The results showed that all life-activity phenotypes were significantly reduced after knocking out *PafS*, whereas all were significantly restored to a similar level to that of MS82 after the complementation of *PafS*. These results suggested that *PafS* plays an important role in the regulation of a range of cellular activities by c-di-GMP in *P. glycinae* MS82.

## 1. Introduction

In the 1990s, the first diguanylate cyclase (DGC) protein gene was identified in *Gluconacetobacter xylinus*, which contains a conserved GGDEF domain [1]. The amino acid sequence of this domain was analyzed, but its biochemical characteristics were not described [2]. Subsequently, Pei and Grishin [3] proposed that the GGDEF domain is the core domain of DGCs. The main rationale for this suggestion is that the GGDEF domain-*PleD* from *Caulobacter crescentus* and *WspR* from *Pseudomonas aeruginosa* can synthesize two molecules of GTP to bis (3′-5′)-cyclic dimeric guanosine monophosphate (c-di-GMP) [4,5]. Subsequently, the GGDEF domain has been a focus of research on the second messenger signaling pathways of bacteria.

As an important domain of DGCs, when it receives different signals from the environment, it can adjust its own concentration or regulate the probability of the binding of the active site containing the GGDEF motif in its domain to the substrate GTP, thus further affecting the synthesis of c-di-GMP [6,7]. In addition, the RXXD motif is located at five amino acid residues upstream of the active center, which is termed the inhibition site (I site). When the synthesized c-di-GMP binds to the I site, the GGDEF domain protein produces non-competitive inhibitor feedback to inhibit the catalytic reaction of DGCs. This feedback regulation can maintain the intracellular c-di-GMP concentration in a dynamic balance [8,9].

The c-di-GMP protein binds to different types of c-di-GMP receptors, and the conjugates participate in and regulate a variety of important bacterial life activities as effector molecules. For example, fluctuation of the c-di-GMP concentration can regulate the initiation of transcription and the post-transcription level of *P. aeruginosa* [10], *Vibrio cholera* [11], *P. fluorescens* [12], *Bacillus cereus* [13], and other bacteria [14], regulate gene expression to varying degrees, the metabolism and production of cellular substances, the cell cycle, differentiation and variation, the stability of ion concentrations, the formation of biofilm, flagella-mediated movement behavior, extracellular polysaccharides (EPS), and cytotoxicity, for example.

Comprising a wide-ranging and adaptable genus of bacteria, *Pseudomonas* spp. are not only pathogenic to plants but also have potential biocontrol effects. Ma [15,16] randomly mutated the *Pseudomonas glycinae* strain MS82 using the EZ-Tn5 transposon system, and obtained a mutant strain MT19 that lost antifungal activity against the mushroom pathogen *Mycogone perniciosa* and *Trichoderma viride*, as a result of mutation of the *PafR* gene containing the GGDEF domain. To explore the signaling mechanism of the genes containing the GGDEF domain in *P. glycinae*, in this study another gene *PafS* containing the same GGDEF domain and some other different domains was selected, and the effects of gene *PafS* on the life activities were investigated through gene knockout and complementation.

## 2. Materials and Methods

### 2.1. Strains, Plasmids, and Culture Conditions

Strains and plasmids used in this study are listed in Table 1. Bacteria were routinely cultured with liquid Lysogenic Broth (LB) medium in a test tube or on LB medium solidified with 1% agar. The concentration of ampicillin (Ap) and gentamicin (Gm) was 50 ng/mL in the different media. The strain MS82 and its mutants were grown in LB medium for determination of biofilm formation and polysaccharide content.

### 2.2. Deletion and Complementation of PafS

In order to understand whether the presence or absence of gene *PafS* will affect the life activity of the strain MS82, construction of the deletion mutant and reversion mutant used the same method of homologous recombination and parental combination as described previously [18]. The upstream and downstream fragments of *PafS* were amplified separately using the primer pairs Δ*PafS*F1/R1 and Δ*PafS*F2/R2 (see Table 2). The plasmid pEX18-Δ*PafS* was generated by inserting both fragments by digestion with the selected restriction enzyme and ligated with the pEX18GM vector by the T_4_ DNA ligase. The plasmid was transformed into *Escherichia coli* strain DH5α and the transformation was confirmed by blue-white spot screening. The plasmid pEX18-Δ*PafS* was purified from the white clones and transformed into *E. coli* strain S17-1λ. The blue clones harboring pEX18-*ΔPafS* were conjugated with *P. glycinae* MS82, and the conjugation was cultured to a protoplast state for homologous recombination at 28 °C on LB plates without antibiotics for 2 days. Recombination occurred on LB medium containing Ap, Gm, and 12.5 mg/mL sucrose overnight at 28 °C to eliminate *E. coli* containing the suicide vector. Clones of the stable deletion mutant Δ*PafS* were able to grow on LB plates containing Ap but not on medium containing Ap and Gm.

For construction of the reversion mutant, several steps differed from those for construction of the deletion mutant as described below. The *PafS* gene containing upstream and downstream fragments was amplified using the primer pair *pafS*-compF/R (see Table 2). The plasmid pEX18-*PafS*-comp was generated by ligation of the *PafS* gene fragment and pEX18GM vector. The blue *E. coli* S17-1λ clones harboring pEX18-*pafS*-comp were conjugated with *ΔPafS*, and the stable reversion mutant *PafS*-comp clones were obtained.

### 2.3. Antifungal Activity Bioassay

The antifungal activities of MS82, Δ*PafS*, and *PafS*-comp were analyzed using the antifungal zone method [19]. Ten microliters of the prepared bacterial solution (approximately 2 × 10^8^ CFU/mL) was spotted in the center of LB solid medium. After the bacterial solution had dried, an appropriate volume of NJ1 spore suspension diluted with sterile water was sprayed with a small watering can (spore concentration is 2 × 10^8^/mL). The inverted plates were incubated in the dark at 28 °C for 2 days, then the diameter of the inhibition zone of each strain was measured. Three replicates of the plate bioassays were performed independently.

### 2.4. Biofilm Formation Assay

The time course of biofilm formation was determined by measuring the ability of cells to grow adherent to the wells of sterile polystyrene microtiter plates (96 flat-bottom wells) as previously described [20]. After the bacterial community is adsorbed and colonizes the surface of the object, the c-di-GMP signaling system mediates the movement of type IV flagella, producing an outermost protective biofilm to resist environmental stimuli. The absorbance of *P. glycinae* is strongest at a wavelength of 590 nm. Therefore, after staining the bacterial liquid with crystal violet, the OD_590_ value was measured with an ultraviolet spectrophotometer to indicate the biofilm formation ability. Plates were incubated under static conditions at 28 °C for up to 48 h. To measure the degree of attachment, cultures were removed from selected wells at 12 h intervals, rinsed with distilled water, stained with 1% crystal violet, and quantified by measuring the OD_590_ of the resulting solution. The experiment was performed four times with three replicates each.

### 2.5. Extracellular Polysaccharides (EPS) Synthesis Assay

Using a modification of a previously described method [21], the strains were incubated in LB liquid medium for 48 h, three volumes of ethanol were added, and the resulting solution was left to stand overnight at 4 °C. The precipitated EPS was collected by centrifugation, dried at 55 °C, and the dry weight was recorded as an estimate of EPS yield. The experiment was repeated three times independently.

### 2.6. Motility Assay

Swimming, swarming, and twitching motility assays were performed as described previously [22]. Semi-solid LB medium was adjusted to contain 0.3% (*w*/*v*) agar for the swimming medium, 0.7% (*w*/*v*) agar for the swarming medium, and 1% (*w*/*v*) agar for the twitching motility medium. Ten microliters of liquid germ of MS82, Δ*PafS*, and *PafS*-comp were inoculated in the center of swimming medium and swarming medium, then incubated at 28 °C for 12 h. Clones were seeded on the bottom of the twitching medium and incubated at 28 °C for 72 h. After incubation, the agar was removed and the bottom of the plate was stained with 0.01% crystal violet for 30 min. The final diffusion diameter was measured for each motility assay, and three replicates were performed.

## 3. Results

### 3.1. Deletion and Complementation of PafS

In the genome of the *P. glycinae* strain MS82 (GenBank accession no. CP028826.1), *PafS* was one of 38 genes that contained the GGDEF domain, and was predicted to be a putative DGC responsible for c-di-GMP synthesis. The *PafS* gene comprised 2124 bp and contained one each of the RESPONSE_R^~^, PAS, GGDEF, and EAL domains.

To explore the potential biological functions of *PafS*, *PafS*-deletion and reversion mutants were constructed in strain MS82. The length of the upstream and downstream fragments of *PafS* were 353 bp and 276 bp, respectively, and a 629 bp fragment was amplified from the *PafS*-deletion mutant *ΔPafS* (Figure 1). However, because some fragments in the designed primer sequence overlapped with the target gene, the amplified band length of MS82 and *PafS*-comp was expected to be 2365 bp (Figure 1). The constructed mutants were verified by PCR and double-enzyme digestion, and the sequences were used as the query for a BLAST search of the NCBI databases. After multiple generations of culture, the mutant strains were genetically stable.

### 3.2. Antifungal Activity Bioassay

The antifungal activity of strain MS82 and its mutants against the pathogen *T. virens* showed variation (Figure 2). The knockout of *PafS* distinctly decreased the antifungal activity of *ΔPafS* (23.50 ± 0.87 mm). In contrast, the antifungal activity of *PafS*-comp did not differ significantly from that of MS82. The deletion and reversion of gene *PafS* resulted in the decrease and increase in the antifungal activity, indicating that *PafS* had a positive regulatory effect on the synthesis of antifungal substances.

### 3.3. Biofilm Formation Assay

The deletion of *PafS* severely affected biofilm formation (Figure 3). After incubation for more than 12 h, the biofilm formation ability of *ΔPafS* was significantly reduced compared with that of MS82 and *PafS*-comp. In contrast, MS82 and *PafS*-comp maintained similar biofilm formation abilities. The results of the biofilm formation assay showed that the gene *PafS* had a positive regulation effect, and the regulation effect increased with the increase in time.

### 3.4. Extracellular Polysaccharides Synthesis Assay

With prolonged incubation, the EPS content of the strains MS82, Δ*PafS*, and *PafS*-comp increased gradually (Figure 4). For each incubation period, the EPS content of Δ*PafS* was significantly lower than that of MS82 and *PafS*-comp. With the knockout and restoration of *PafS*, the EPS content changed from low (0.24 ± 0.01 g, 48 h) to high (0.39 ± 0.01 g, 48 h). Although a slight difference in EPS content was observed between MS82 and *PafS*-comp at each time period, the difference was not significant. Deletion of the gene *PafS* reduced EPS production, suggesting that *PafS* promotes the EPS synthesis of strain MS82. 

### 3.5. Motility Assay

The results of the motility assay were consistent with those for antifungal activity, biofilm formation, and EPS content (Figure 5). The swimming, swarming, and twitching movement abilities of Δ*PafS* differed significantly from those of the wild-type MS82 and the *PafS*-comp mutant. Complementation of *PafS* in the reversion mutant *PafS*-comp restored all three motor performance abilities. Notably, the degree of recovery of the swarming movement was greater than that of the swimming and twitching movements, and the diffusion diameter of *PafS*-comp was slightly larger than that of the wild-type MS82 strain, but the differences were not significant. The regulation of gene *PafS* can significantly promote three kinds of motility.

## 4. Discussion

In a previous study, we used site-directed mutagenesis and random mutagenesis to generate mutations of *PafR*, which showed similar experimental results as observed in the present study [15]. We applied the method of biparental conjugation and homologous recombination to successfully knockout and complement the gene *PafS*, which contains the GGDEF domain, in *P. glycinae* strain MS82. This procedure resulted in the corresponding deletion mutant *ΔPafS* and reversion mutant *PafS*-comp, which showed significant phenotypic differences. Thus, a novel procedure for gene mutation in *P. glycinae* is described here.

The *PafR* gene, which also contains the GGDEF domain, is closely associated with the antifungal activity of *P. glycinae* MS82 [15]. In the present study, the antifungal activity bioassay showed that the antifungal-activity phenotype differed significantly before and after the deletion of the *PafS* gene containing the GGDEF domain, and that the wild-type phenotype was restored after complementation of the gene. We speculate that the deletion of *PafS* affects the expression of certain metabolic genes in the MS82 strain, which blocks the recognition of specific promoters by the DGC core enzymes, thereby inhibiting the secretion of antibiotic metabolites inhibitory to pathogenic fungi. These substances mainly include the antibiotics pyoverdine (Pvd), 2,4-diacetylphloroxylic (DAPG), phenazine-1-carboxylic acid (PCA), pyolueorin (PLT), and pyrrolnitrin (PRN) [23,24]. Both mutants of gene *PafR* and *PafS* showed significantly decreased antifungal activities, suggesting that their deletion would reduce the synthesis of one or more antifungal substances in *P. glycinae* MS82. These antifungal substances may combine their specificity. For example, the rhizosphere bacterium *P. fluorescens* promotes the growth and improves the autoimmunity of the host plant towards the green mold disease of citrus, and thus acts as a biocontrol agent to effectively inhibit the incidence of plant diseases.

During biofilm formation, EPS synthesis and the movement behavior of bacteria are closely associated. Motility is determined by external structures, such as flagella and pili, on the bacterium surface. Polysaccharides form the “skeleton” of the biofilm and the composition varies with the formation of the biofilm. The first adhesion stage of bacterial biofilm formation begins with the swimming movement mediated by type IV flagella [25]. Through the secretion of various glycoproteins, mucopolysaccharides, metal ions, and other components, the bacteria adhere to the surface of an object within a short period to form a multicellular aggregation embedded within a self-produced matrix, which continues to play an important role in the selective transportation of nutrients and in the catalysis of metabolic processes. At its maturity, the biofilm is regulated by the synthesis and polymerization of extracellular polysaccharide synthesis receptor proteins. For example, in *P. aeruginosa* PAO1, the *FleQ* gene affects the adhesion of the bacteria by inhibiting flagella synthesis [26]. The FleQ protein binds to the promoter regions of the Pel operon and Psl operon to transcriptionally regulate the biosynthesis of the Pel and Psl polysaccharides. In addition, FleQ regulates the synthesis of bacterial extracellular polysaccharides by affecting the activity of the corresponding glycosyltransferase [27]. These regulatory mechanisms occur in the second signaling system pathway of c-di-GMP. Therefore, as the core site of c-di-GMP synthase DGCs, the GGDEF domain directly affects bacterial adhesion [3], EPS synthesis, and type IV flagella-mediated motility. In the present study, the phenotypes of biofilm formation, EPS synthesis, and motility were reduced after the deletion of *PafS*, whereas the corresponding life activities were restored after complementation with *PafS*. These results are consistent with the above findings, which confirm that GGDEF domain-containing genes influence the function of c-di-GMP synthase DGCs, and further affect the c-di-GMP signaling pathway to regulate biofilm formation, EPS content, and motility.

Although we confirmed that the function of the GGDEF-domain-containing gene *PafS* in *P. glycinae* MS82 is consistent with previous findings, owing to the diversity of the c-di-GMP signaling pathway and the corresponding receptor proteins between and within populations, the GGDEF domain of different genomes distributed in different populations will differ in the degree and direction of regulation [28]. For example, in *Pseudomonas* spp., the GGDEF-domain-containing *WspR* gene in *P. fluorescens* Pf0-1 has been mutated to produce a weak positive increase in biofilm formation without affecting the corresponding swimming motion [18,29]. The same mutation in *P. fluorescens* F113 exhibits enhanced swimming motility [30]. In *P. aeruginosa* PAO1, the mutation of *wspR* inhibits bacterial motility [31]. The comparison of the present results for biofilm formation, EPS content, and motility with the above-mentioned findings indicates that the genes containing the GGDEF domain in *Pseudomonas* spp. strains perform a similar regulatory function. However, the regulatory functions of genes containing the GGDEF domain at different positions in the genome in different strains also differ. With regard to motility, the present results differed from those for *P. fluorescens* F113 [32], but were consistent with the experimental results reported for *P. aeruginosa* PAO1 [33].

The present study represents the first phase in exploring the c-di-GMP signaling mechanism in *P. glycinae* MS82. However, further research on the signaling pathway is required to assess the roles of the remaining GGDEF domains in MS82 to determine whether their influence on bacterial life activities is consistent, to identify homologous genes in other genomes, to detect commonalities of c-di-GMP in MS82 and other *Pseudomonas* strains, and to elucidate the complete c-di-GMP signaling pathway.

## Figures and Tables

**Figure 1 microorganisms-10-02342-f001:**
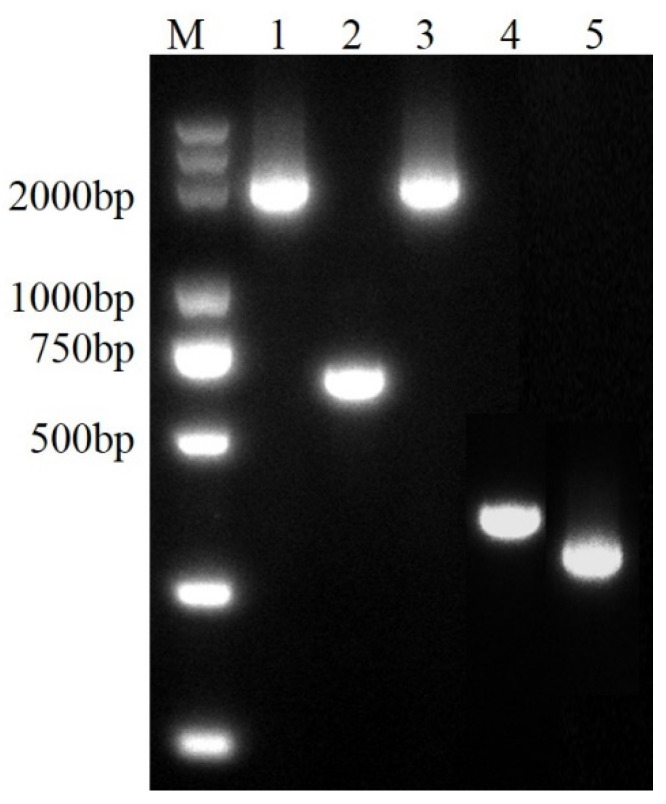
**The** PCR confirmation of *PafS*-deletion and reversion mutants of *P. glycinae* strain MS82. M, marker 5000; 1, wild-type MS82 strain with the primer pair *pafS*-compF/R; 2, deletion mutant Δ*PafS* with the primer pair Δ*PafS*-F1/R2; 3, reversion mutant *PafS*-comp with the primer pair *pafS*-compF/R; 4, the upstream fragment of *PafS* with the primer pair Δ*PafS*-F1/R1; 5, the downstream fragment of *PafS* with the primer pair *ΔPafS*-F2/R2.

**Figure 2 microorganisms-10-02342-f002:**
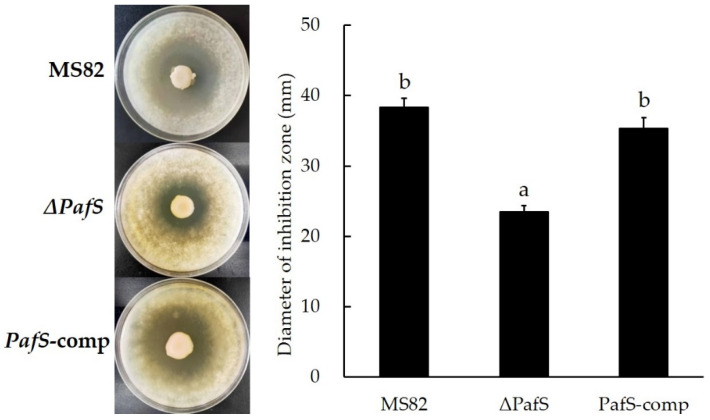
Inhibition of *T. virens* by *PafS*-deletion and reversion mutants of *P. glycinae* strain MS82. Different lowercase letters above bars indicate a significant difference compared with the control (*p* < 0.05). The error bars indicate the SD and the statistical test used Student’s *t*-test.

**Figure 3 microorganisms-10-02342-f003:**
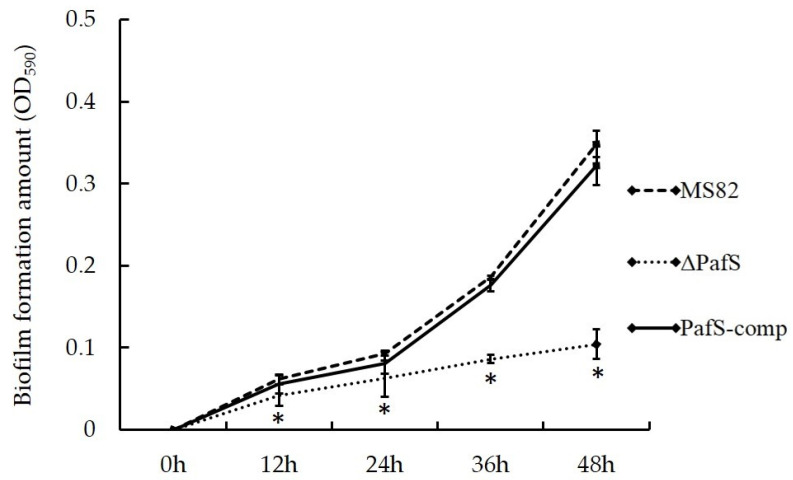
Biofilm formation by *PafS*-deletion and reversion mutants of *P. glycinae* strain MS82. ‘*’ under bars indicate a significant difference compared with the control (*p* < 0.05). The error bars indicate the SD value.

**Figure 4 microorganisms-10-02342-f004:**
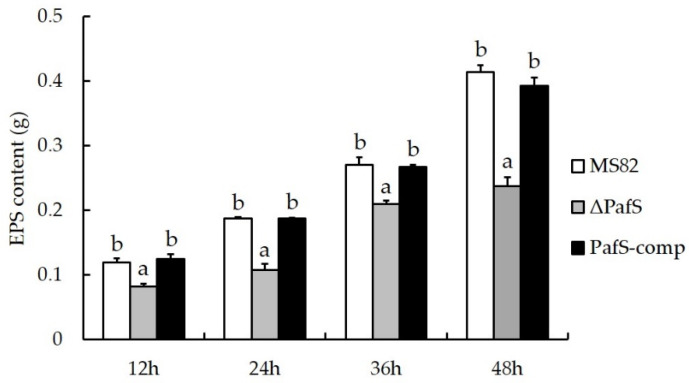
Extracellular polysaccharides (EPS) content of *PafS*-deletion and reversion mutants of *P. glycinae* strain MS82. Different lowercase letters above bars for a specific timepoint indicate a significant difference compared with the control (*p* < 0.05). The error bars indicate the SD and the statistical test used Student’s *t*-test.

**Figure 5 microorganisms-10-02342-f005:**
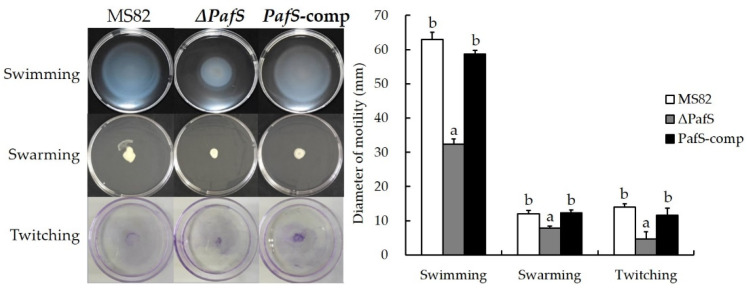
Motility assay of *PafS*-deletion and reversion mutants of *P. glycinae* strain MS82. Different lowercase letters above bars for a motility type indicate a significant difference compared with the control (*p* < 0.05). The error bars indicate the SD and the statistical test used Student’s *t*-test.

**Table 1 microorganisms-10-02342-t001:** Bacterial strains and plasmids used in this study.

Strain or Plasmid	Relevant Characteristics	Source
	***Pseudomonas glycinae*** [17]	
MS82	Wild-type strain, 28 °C, Ap^r 1^	This study
Δ*PafS*	MS82 with deletion of *PafS*, 28 °C, Ap^r^	This study
* PafS * -comp	MS82 with reverse of *PafS*, 28 °C, Ap^r^	This study
	** *Escherichia coli* **	
DH5α	*supE*44 Δ*lac*U169 (*Φ*80 *lacZ*ΔM15) *hsdR*17 *recA*1 *endA*1 *gyrA*96 *thi*-1 *relA*1, 37 °C	TSINGKE, Beijing, China
S17-1λ	*RP*4-2(Km::Tn7,Tc::Mu-1), *pro*-82, *LAMpir*, *recA*1, *endA*1, *thiE*1, *hsdR*17, *creC*510, 37 °C	WEIDI, Shanghai, China
	** *Trichoderma virens* **	
NJ1	A pathogen isolated from mushroom culture medium	This laboratory
	**Plasmids**	
pEX18GM	Suicide vector, *SacB*, Gm^r 2^	FENGHUI, Changsha, China
pEX18-*ΔPafS*	PEX18 with upstream and downstream fragments of *PafS*, Gm^r^	This study
pEX18-*PafS*-comp	PEX18 with *PafS* gene containing upstream and downstream fragments, Gm^r^	This study

^1^ Ap^r^, ampicillin resistance; ^2^ Gm^r^, gentamicin resistance.

**Table 2 microorganisms-10-02342-t002:** Primers used in this study.

Primer	Sequence 5′→3′	Enzyme	Annealing Temperature
ΔPafS-F1 ΔPafS-R1	CCAAGCTTGGTAAAGCGTGCGGTGTCCT ^1^ GGGGTACCCGATCAACCCGGACGAGACT	HindIII KpnI	55 °C
ΔPafS-F2 ΔPafS-R2	GGGGTACCATGCTGCTTCTCTTGTCGGG GCTCTAGAAAGCGTAGAGGGATTTTTTG	KpnI XbaI	55 °C
PafS-compF PafS-compR	ACGGCCAGTGCCAAGCTTGGTAAAGCGTGCGGTGTC GGTACCCGGGGATCCTCTAGACGTTGATGACTACCCTGAAAATCT	HindIII XbaI	65 °C

^1^ The underlined sequence is the restriction enzyme site.

## Data Availability

Not applicable.

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
