# Peer review of "PafS* Containing GGDEF-Domain Regulates Life Activities of *Pseudomonas glycinae* MS82"

_microorganisms, 2022, doi:10.3390/microorganisms10122342_

Round 1
Reviewer 1 Report
In this manuscript, the authors demonstrated the role of PafS on the phenotype of the bacterium Pseudomonas glycinaestrain MS82. PafS is part of the GGDEF-domain of diguanylate cyclase (DGC), which is involved in the synthesis of cyclic dimeric guanosine monophosphate (c-di-GMP). By constructing knockout mutant and complemented isolates, the authors showed that PafS has a role in the antifungal activity, biofilm formation ability, extracellular polysaccharide content, and the motor behavior of strain MS82.
The study is well-designed and straight forward and the results are easily interpreted. However, this study only characterized one gene of 38 genes containing GGDEF domain. Previously, the same group found a similar result with pafR, another gene of the GGDEF domain. There are other studies that also demonstrate the gene-specific role for genes in the GGDEF domain in Pseudomonas species. What this study is lacking is the mechanism of PafS to alter organisms behavior. However, the findings of this study are important to understand the role of the GGDEF domain in the regulation of c-di-GMP in organisms.
I have some suggestions and comments to improve the quality of the manuscript.
Specific comment:
The authors should mention the reversion mutant strain Delta5092 as the pafS complement strain throughout the manuscript. The naming of the strains is also confusing, as the Delta symbol should be used for the mutant strain and not for the complemented strain. To make it simple for all readers, the authors can consider naming the mutant strain as DeltapafS and the complemented strain as pafS-comp.
Title: Please give the organism name in the title of the manuscript.
Introduction: The introduction section should be expanded with the already characterized GGDEF genes in Pseudomonas. In addition, the authors should include the context of the PafS, and why they thought this gene is important for the bacterium phenotype.
Results: Overall results section can be described in more detail. Currently, it is very brief and does not depict a story. e.g., which experiment was done first and why? What do the differences in phenotypes mean? What is the statistical data and what it means? Finally, what is the overall summary of each of the experiments?
Figures: The resolution of the figures in the manuscript is not good and not clear to visualize. All figure legends should mention the number of times the experiment was conducted.
Figure 1: Please include a separate panel showing the schematic diagram of the pafS and upstream and downstream genes of DNA fragments for the wt, mutant, and complemented isolates (with isolate name/ID). Also please show the primer location, IDs, and the direction of the primers.
Figure 2: Instead of a bar plot, please use a box plot to show the individual values, median, and standard deviation.
Figure 3: Please show the statistical significance for each of the time points.
Figure 4: Please use a box plot to show the individual values, median, and standard deviation.
Table 1: Are TSINGKE, WEIDI, and FENGHUI the name of the companies? If so, please add the city and country name.
Line 58: Provide the full name of the bacterium when it first appeared.
Line 62: The sentence ‘To explore the signaling mechanism of P. glycinae, in this study the functional effects of GGDEF domain-containing genes in the c-di-GMP signaling pathway of P. glycinae MS82 on bacterial life activities were investigated.’ should be modified as this study analyzed only one gene.
Line 71-73: Cultures for motility is best suited in the section of Motility assay in Methods.
Line 81: Name the enzymes.
Line 121: Give the full name of EPS in the title.
Line 127-134: Please describe the measurement for each of the assays (swimming, swarming, and twitching) separately.
Line 147: Name the enzymes.
Line 241: Give the reference(s).
Line 253: Mutation of what? Which genes?
Line 259-260: Did these studies (references # 31 and 32) show the results for the same gene?
Line 261: Why do you think that it is the first step?
Please mention whether there is any relation between the PafS and PafR. Are these two-component response regulators? Did you name these genes/proteins?
Author Response
First of all, thank you very much for your valuable suggestions on the revision of my manuscript. I have carefully studied your suggestions for me, and have revised most parts according to your suggestions, some of which I have explained. Now I will make a detailed explanation for each of your suggestions.
The authors should mention the reversion mutant strain Delta5092 as the pafS complement strain throughout the manuscript. The naming of the strains is also confusing, as the Delta symbol should be used for the mutant strain and not for the complemented strain. To make it simple for all readers, the authors can consider naming the mutant strain as DeltapafS and the complemented strain as pafS-comp.
I have revised it according to the advice.
Title: Please give the organism name in the title of the manuscript.
I have revised it according to the advice.
Introduction: The introduction section should be expanded with the already characterized GGDEF genes in Pseudomonas. In addition, the authors should include the context of the PafS, and why they thought this gene is important for the bacterium phenotype.
I have revised it according to the advice.
Results: Overall results section can be described in more detail. Currently, it is very brief and does not depict a story. e.g., which experiment was done first and why? What do the differences in phenotypes mean? What is the statistical data and what it means? Finally, what is the overall summary of each of the experiments?
I have revised it according to the advice.
Figures: The resolution of the figures in the manuscript is not good and not clear to visualize. All figure legends should mention the number of times the experiment was conducted.
I have revised it according to the advice.
Figure 1: Please include a separate panel showing the schematic diagram of the pafS and upstream and downstream genes of DNA fragments for the wt, mutant, and complemented isolates (with isolate name/ID). Also please show the primer location, IDs, and the direction of the primers.
I have added the separate panel showing the schematic diagram of the upstream and downstream pafS of DNA fragments for the wt. But in the paper we did not design the primers for the gene pafS. The primer location, IDs, and the direction of the primers have showed in the table 2.
Figure 2: Instead of a bar plot, please use a box plot to show the individual values, median, and standard deviation.
I tried to change it to a box plot, but it didn't work very well, so I kept the bar plot. I feel that the bar plot can also express the same experimental results.
Figure 3: Please show the statistical significance for each of the time points.
I have revised it according to the advice.
Figure 4: Please use a box plot to show the individual values, median, and standard deviation.
I tried to change it to a box plot, but it didn't work very well, so I kept the bar plot. I feel that the bar plot can also express the same experimental results.
Table 1: Are TSINGKE, WEIDI, and FENGHUI the name of the companies? If so, please add the city and country name.
I have revised it according to the advice.
Line 58: Provide the full name of the bacterium when it first appeared.
I have revised it according to the advice.
Line 62: The sentence ‘To explore the signaling mechanism of P. glycinae, in this study the functional effects of GGDEF domain-containing genes in the c-di-GMP signaling pathway of P. glycinae MS82 on bacterial life activities were investigated.’ should be modified as this study analyzed only one gene.
I have revised it according to the advice.
Line 71-73: Cultures for motility is best suited in the section of Motility assay in Methods.
I have revised it according to the advice.
Line 81: Name the enzymes.
The Name the enzymes have showed in table 2.
Line 121: Give the full name of EPS in the title.
I have revised it according to the advice.
Line 127-134: Please describe the measurement for each of the assays (swimming, swarming, and twitching) separately.
The measurement for the assays were the same as the reference, so I did not write again.
Line 147: Name the enzymes.
The Name the enzymes have showed in table 2.
Line 241: Give the reference(s).
Refer to the results of the previous literature research in this paragraph
Line 253: Mutation of what? Which genes?
I have revised it according to the advice.
Line 259-260: Did these studies (references # 31 and 32) show the results for the same gene?
Those are different genes.
Line 261: Why do you think that it is the first step?
Please mention whether there is any relation between the PafS and PafR. Are these two-component response regulators? Did you name these genes/proteins?
I have revised it according to the advice.
Reviewer 2 Report
In this study, the authors investigated the functional effects of GGDEF domain-containing genes in the c-di-GMP signaling pathway of P. glycinae MS82 on bacterial life activities. The article covers an exciting research topic and valuable information that can interest Journal readers. The overall results and discussion are clear and well defined, supported by relevant literature review, however, there are some questions that need to be addressed for more clarification. The manuscript may be accepted after the following inputs:
1. In materials and methods (66-74), Table 1: Source of mutation not clearly mentioned. Please provide the source. Is it natural mutation or induced?
2. Line No. 77-80: Please clarify the effect of mutant deletion.
3. 214-217: Please clarify the status/level of antifungal compounds in mutants i. e. increase or decrease and its application.
4. Line No. 237-238: The role of HSP 54 & 28 is not clearly discussed. They have important functions in flagella movement. Please describe it in brief.
5. Line no. 261-266: Please write a separate conclusion section and elaborate on the important conclusions drawn and the study's future scope.
Author Response
First of all, thank you very much for your valuable suggestions on the revision of my manuscript. I have carefully studied your suggestions for me, and have revised most parts according to your suggestions, some of which I have explained. Now I will make a detailed explanation for each of your suggestions.
- In materials and methods (66-74), Table 1: Source of mutation not clearly mentioned. Please provide the source. Is it natural mutation or induced?
The method of 2.2 is the forming process of mutants.
- Line No. 77-80: Please clarify the effect of mutant deletion.
I have revised it according to the advice.
- 214-217: Please clarify the status/level of antifungal compounds in mutants i. e. increase or decrease and its application.
I have revised it according to the advice.
- Line No. 237-238: The role of HSP 54 & 28 is not clearly discussed. They have important functions in flagella movement. Please describe it in brief.
I am so sorry that I am not clearly about HSP 54 & 28.
- Line no. 261-266: Please write a separate conclusion section and elaborate on the important conclusions drawn and the study's future scope.
I have write the conclusion section in the above paragraph. The point of this paragraph is the study's future scope.
Reviewer 3 Report
The manuscript explored the role of the gene PafS which contains a GGDEF domain. They have shown that this gene regulates various cellular activities by c-di-GMP. The deletion of this gene reduces the major life activities of the bacteria i.e., biofilm formation, antifungal activity etc. Their results confirm the importance pf GGDEF domain in the cellular functions of the bacterium. The experiments are well performed, and conclusions are convincing. I believe to prove that the domain is essential for the activity of this gene it would be more appropriate to delete just the domain from the gene and then see the impact on the phenotype. The presentation of the manuscript needs more work. There are grammatical and typo mistakes which needs to be addressed.
Comments:
1. Instead of saying bacterium it would be better to add the organism’s name in the title.
2. Line 68 Joseph and Michael not in the references.
3. Line 156 knockout of PafS decreased the antifungal activity instead of increased.
4. Line 166 increasingly reduced does not sound right in this sentence, authors should use significantly reduced.
Author Response
First of all, thank you very much for your valuable suggestions on the revision of my manuscript. I have carefully studied your suggestions for me, and have revised all four parts according to your suggestions.
Round 2
Reviewer 3 Report
The authors have addressed the issues raised in the first round and I believe the present form is suitable for publication.